# Cost-effectiveness evaluations of the 9-Valent human papillomavirus (HPV) vaccine: Evidence from a systematic review

Rashidul Alam Mahumud[1,2,3,4,5]*, Khorshed Alam[1,2], Syed Afroz Keramat[1,2,6], Gail M. Ormsby[7], Jeff Dunn[1,8,9], Jeff Gow[1,2,10]

**1** Health Economics and Policy Research, Centre for Health Research, University of Southern Queensland, Toowoomba, Queensland, Australia, **2** School of Commerce, University of Southern Queensland, Toowoomba, Queensland, Australia, **3** Health Systems and Population Studies Division, Health Economics and Financing Research, International Centre for Diarrhoeal Disease Research, Bangladesh (icddr,b), Dhaka, Bangladesh, **4** School of Social Sciences, Western Sydney University, Penrith, Australia, **5** Translational Health Research Institute, Western Sydney University, Penrith, Australia, **6** Economics Discipline, School of Social Science, University of Khulna, Khulna, Bangladesh, **7** School of Health and Wellbeing, University of Southern Queensland, Toowoomba, Queensland, Australia, **8** Cancer Research Centre, Cancer Council Queensland, Fortitude Valley, Queensland, Australia, **9** Prostate Cancer Foundation of Australia, St Leonards, NSW, Australia, **10** School of Accounting, Economics and Finance, University of KwaZulu-Natal, Durban, South Africa

\* rashidul.icddrb@gmail.com, rashed.mahumud@usq.edu.au

**Data Availability Statement:** All relevant data are within the manuscript and its Supporting Information files.

## Abstract

### Introduction

The World Health Organization (*WHO*) recommends that human papillomavirus (*HPV*) vaccination programs are established to be cost-effective before implementation. WHO recommends HPV vaccination for girls aged 9–13 years to tackle the high burden of cervical cancer. This review examined the existing evidence on the cost-effectiveness of the 9-valent HPV vaccine within a global context.

### Methods

The literature search covering a period of January 2000 to 31 July 2019 was conducted in *PubMed* and *Scopus* bibliographic databases. A combined checklist (i.e., *WHO*, Drummond and *CHEERS*) was used to examine the quality of eligible studies. A total of 12 studies were eligible for this review and most of them were conducted in developed countries.

### Results

Despite some heterogeneity in approaches to measure cost-effectiveness, ten studies concluded that *9vHPV* vaccination was cost-effective and two did not. The addition of adolescent boys into immunisation programs was cost effective when vaccine price and coverage was comparatively low. When vaccination coverage for females was more than 75%, gender neutral *HPV* vaccination was less cost-effective than vaccination targeting only girls aged 9–18 years. Multi cohort immunization approach was found cost-effective in the age range of 9–14 years. However, the upper age limit at which vaccination was found not cost-

**Funding:** The author(s) received no specific funding for this work.

**Competing interests:** The authors have declared that no competing interests exist.

effective requires further evaluation. This review identified duration of vaccine protection, time horizon, vaccine price, coverage, healthcare costs, efficacy and discounting rates as the most dominating parameters in determining cost-effectiveness.

## Conclusions

These findings have implications in extending *HPV* immunization programs whether switching to the 9-valent vaccine or the inclusion of adolescent boys' vaccination or extending the age of vaccination. Further, this review also supports extending vaccination programs to low-resource settings where vaccine prices are competitive, donor funding is available, burden of cervical cancer is high and screening options are limited.

## Introduction

Cervical cancer (*CC*) is the third most common cancer and the leading cause of cancer-related deaths in women worldwide [1]. Approximately 570,000 new cases of *CC* were diagnosed in 2018, composing 6.6% of all cancers in women [1]. The burden of *CC* is an alarming issue across the globe, especially in low-and middle-income countries (*LMICs*). Approximately 85% of *CC* cases and 90% of deaths from *CC* occur in *LMICs* [1]. Persistent infections with human papillomavirus (*HPV*) are a key cause of *CC* and is an established carcinogen of *CC* [2]. *HPV* is predominantly transmitted to women of reproductive age through sexual contact [3]. Most *HPV* infections are transient and can be cleared up within a short period, usually a few months after their acquisition. However, untreated *HPV* infections can continue and evolve into cancer in some cases. There are more than 100 types of *HPV* infections, and high-risk types develop into *CC* [4]. Thirteen high-risk *HPV* genotypes are known to be predominantly responsible for malignant and premalignant lesions of the anogenital area [5], and these are the leading causes of most aggressive *CC* [6]. Further, *HPV* is also responsible for the majority of anogenital cervical cancers, including anal cancers (88%), vulvar cancers (43%), invasive vaginal carcinomas (70%), and all penile cancers (50%) globally [4].

The burden of *CC* (i.e., high incidence and mortality rates) globally is preventable through the implementation of a primary prevention strategy such as vaccination [1]. There are vaccines that can protect common cancer-causing types of *HPV* and reduce the risk of CC significantly. Three types of *HPV* vaccines, namely *bivalent* (Cervarix), *quadrivalent* (Gardasil) and 9-valent vaccine (Gardasil-9), are currently available in the market. Unfortunately, as of March 2017, only 71 countries (37% of all countries) have included *HPV* vaccines in their national immunization programs for girls, and 11 countries (6%) included for both sexes [2]. The first global recommendation on *HPV* vaccination was proposed by the World Health Organization's Strategic Advisory Group of Experts on Immunization in October 2008 [7], where *HPV* vaccination was recommended for girls aged 9–13 years. This recommendation was updated in April 2014 [8], with the emphasis to include extended 2-dose *HPV* immunization for girls aged 9–14 years, who were not immune compromised. With the recent licensing of the 9-valent vaccine and the introduction of various *HPV* vaccination strategies, an update on the current recommendations of *HPV* vaccination are inevitable. The goals of the immunisation program are to combat the acquisition and spread of *HPV* infections, and achieving optimum coverage through effective delivery systems. According to the underlying distribution of *HPV* infection types of *CC*, the *9vHPV* vaccine builds population-level strong immunity against *HPV*-6, 11, 16, 18, 31, 33, 45, 52, and 58 infections [5] that cumulatively contributed

approximately 89% of all *CCs* globally [9]. With respect to the primary prevention of *HPV* infection, it is expected that the *9vHPV* vaccine can reduce the lifetime risk of diagnosis with *CC* by an additional 10% in immunised cohorts compared with the *4vHPV* vaccine and by an additional 52% in non-vaccinated cohorts [10].

This review aims to update the current evidence on the economic viability of *HPV* vaccination. In addition, this study aims to examine the cost-effectiveness of the 9-valent vaccine when boys are included and when age cohorts are varied, from the global context. This review may be used as comprehensive evidence of general trends on the ongoing cost-effectiveness evaluation of *HPV* vaccine.

## Materials and methods

### Study design

Published original academic literature that examined the cost-effectiveness of *9vHPV* vaccination were included in this systematic review. A wide type of study perspectives including societal and health systems perspectives were employed. A search strategy was adopted considering all countries regardless of perspective or vaccine delivery strategy. A combined *WHO* [11], Drummond [12] and *CHEERS* [13] checklist was used to evaluate the quality of included studies.

### Search strategy and sources

A literature search for the period of January 2000 to 31 July 2019 was conducted using *PubMed* and *Scopus* bibliographic databases. This study searched for articles with no language restrictions. The literature search was performed by searching Scopus and PubMed databases to identify relevant articles following the inclusion criteria. Search inclusion terms included 'economic evaluation', 'cost-effectiveness', 'analysis', 'human papillomavirus', '*HPV*', 'vaccine', 'vaccinated', 'vaccination', 'cervical cancer', 'non-valent', '9 or nine-valent' (Appendix A). Reference lists for selected studies were checked to identify relevant studies for inclusion.

### Study selection

Three authors (*RAM*, *SAK and GMO*) of the review team independently examined the titles and abstracts of the articles that met the selection criteria. The existing academic literature in the cost-effectiveness of *9vHPV* vaccination was searched. Language restrictions were not applied. The eligibility of studies for inclusion was determined following a three-stage screening process. The first stage involved screening studies by title to eliminate duplicates. The second stage required the reading of abstracts to determine their relevance to this study. The third stage necessitated the reading of full texts of the retained studies as reflected in Fig 1. *RAM* carried out and recorded the above process, and shared the record with *SAK and GMO* for verification. Discrepancies were discussed and resolved by consensus.

### Data checking

The study strategy followed a number of checks to ensure consistency of approach, including a discussion about discrepancies within the study team. For each outcome and model input parameters, the authors identified the proportion of missing observations. Datasets were combined to form a new master dataset where model input assumptions and outcome-related parameters used in the original studies were included. Further, three authors independently assessed the analytical quality of the preliminary selected studies using appropriate tools for

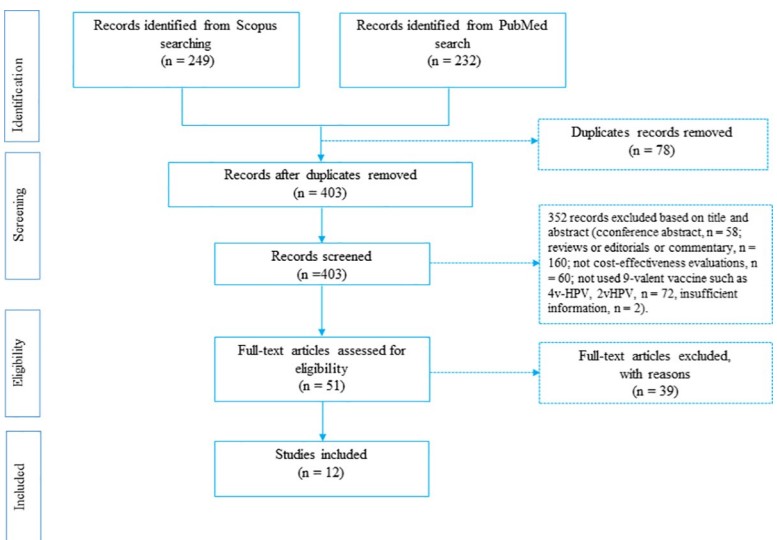

**Fig 1. PRISMA flow-chart for systematic review of studies.**

examining risk of bias. Disagreements on inclusions were resolved by discussion with a third review author.

## Data extraction (selection and coding)

The study selection process was conducted following the *PRISMA* guidelines [14]. Data were extracted to develop a comprehensive data matrix which summarises the study characteristics such as authors, settings, perspective, threshold, outcome-related parameters and other necessary information.

## Strategy for data synthesis

Three authors (*RAM*, *SAK and GMO*) independently reviewed the titles and abstract. Data from all eligible studies were extracted by the same two authors using a standardized data collection form. A matrix was developed to summarise the characteristics and findings of the studies. Studies were characterized by incorporating four themes: (i) study used *9vHPV* vaccine to examine the cost-effectiveness, (ii) target population demographic characteristics (e.g., gender-neutral and multiple age cohort immunisation), (iii) study perspectives, model and economic level of each country, and (iv) model input and outcome-related parameters.

To compare findings across the selected studies, incremental cost-effectiveness ratios (*ICERs*) and standardized cost-effectiveness were outlined. In terms of standardized cost-effectiveness scenarios, these studies used the heuristic cost-effectiveness threshold guided by the *WHO* [15], wherein an intervention or program was evaluated to be cost-effective if the *ICER/DALYs* averted was less than three times a country's annual per capita Gross Domestic Product (*GDP*). Further, the *WHO* constructed three broad decision rules: (i) an intervention or program was recommended as very cost-effective if *ICER/DALYs* averted <1 time *GDP* threshold; (ii) cost-effective if *ICER/DALYs* averted ≥ 1 time *GDP* threshold and ≤ 3 times *GDP* threshold; and (iii) not cost-effective if *ICER/DALYs* averted >3 times *GDP* threshold [16]. Examining whether an *ICER* offered by any strategy signifies value for money requires comparison to a cost-effectiveness threshold (*CET*). The *CET* refers to the health effects foregone (i.e., opportunity costs) related to resources being devoted to an intervention and consequentially being

unavailable for other health-care priorities. Policy makers should be willing to invest their limited resources in the strategy offering the greatest health gains. The review may serve as an important evidence with respect to methodological and current practices of cost-effectiveness evaluation studies such as determination of study research questions; the study perspective adopted, the duration of vaccine protection, time horizon and discount rate; explanation of model performed for data analysis; model input assumptions behind the estimation of associated costs and outcome parameters; reporting of *ICERs*; most dominant parameters of sensitivity analysis; examination of study conclusions and recommendations as well as financial disclosure of the selected studies.

## Study characteristics

Four hundred and eighty-one articles were yielded through the primary search, of which 78 articles were discarded because of duplication. Fifty-one articles were considered for full-text review after screening by title and abstract. Of these, 12 articles were eligible for the final review (Table 1). Three hundred fifty-two articles were excluded from this study following the inclusion criteria. The reasons for exclusion were: conference abstract (n = 58), reviews or editorials or commentary (n = 160), not cost-effectiveness evaluations (n = 60), did not use 9-valent vaccine (*4v-HPV*, *2v-HPV*; n = 72) and insufficient information (n = 2). Finally, 12 articles were included in this review (Fig 1).

## Settings and funding

Single country studies mostly focused on high-income settings [4,17,26–28,18–25] (Table 2). However, a single study was found that covered two low-income countries (e.g., Kenya and Uganda) [29]. Eight studies were funded by research organisations [4,17,19,21–24,29], while two studies did not state funding sources [20,27]. The Bill and Melinda Gates Foundation was the sole funder of one study [29] and three studies were funded by the Centre for Disease Control (*CDC*) [21,24,27]. Further, five studies were conducted in United States [20,22,24,27,28], one study was conducted in each of Germany [23], Italy [4], China [18], Australia [25], Austria [17], and Canada [19]. Low resource countries mostly depend on external funding agency for *HPV* vaccine programs, hence these countries may have less impetus for cost-effectiveness studies to inform local decision making as priorities are driven by external considerations.

## Study questions

Most studies (8 out of 12 studies) investigated the cost-effectiveness of introducing *9vHPV* vaccination to preadolescent girls aged 12 or younger [4,17–19,22,24–26,29]. Four studies assessed vaccinating 12 years or older girls [20,23,27,28]. All studies investigated vaccination either as an addition to existing screening programs or (more commonly) as opportunistic preventive programs or none at all. Further, most studies considered a range of vaccination and screening options to find the most cost-effective combination.

## Analytical model

Nine studies used a dynamic economic model for examining the cost-effectiveness of *HPV* vaccination programs [4,17,27,28,18,20–26], two studies used a static model [19,29], and one study used a Markov model for analytical exploration [18] (Table 2). However, some studies did not explicitly account for the pathologic transition from *HPV*-acquisition to *HPV*-associated disease [4,18,25,27,28], pathologic transition [4,23] and herd immunity [17,19,20,24,27,28].

**Table 1. Characteristics of twelve included cost-effectiveness studies of *9vHPV* vaccination.**

| Characteristics | Number of studies (n) | Percentage (%) |
|---|---|---|
| Selected articles | 12 | 100 |
| Year of publication | | |
| *2014* | 2 | 17 |
| *2016* | 7 | 58 |
| *2017* | 2 | 17 |
| *2018* | 1 | 8 |
| Name of Journal | | |
| *BMC Infectious Diseases* | 2 | 17 |
| *Cost Effectiveness and Resource Allocation* | 1 | 8 |
| *Expert Review of Pharmacoeconomics & Outcomes Research* | 1 | 8 |
| *Human Vaccines & Immunotherapeutics* | 1 | 8 |
| *International Journal of Cancer* | 1 | 8 |
| *The Lancet Public Health* | 1 | 8 |
| *PLoS ONE* | 1 | 8 |
| *The Journal of Infectious Diseases* | 2 | 17 |
| *Vaccine* | 1 | 8 |
| *Journal of the National Cancer Institute* | 1 | 8 |
| Study setting | | |
| *Australia* | 1 | 8 |
| *Austria* | 1 | 8 |
| *Canada* | 1 | 8 |
| *China* | 1 | 8 |
| *Germany* | 1 | 8 |
| *Italy* | 1 | 8 |
| *Kenya and Uganda* | 1 | 8 |
| *United States* | 5 | 42 |
| Main location of first author | | |
| *Research institute* | 8 | 67 |
| *Research group* | 1 | 8 |
| *Hospital or University* | 3 | 25 |
| Conflict of interest | | |
| *Yes* | 6 | 50 |
| *No* | 6 | 50 |

## Thresholds and perspectives

In terms of the cost-effectiveness scenario, four studies used the heuristic cost-effectiveness threshold proposed by the *WHO*. These studies used either one or three times *GDP* per capita [18,19,24,29]. The majority of studies adopted local thresholds (e.g., willingness to pay) while three studies considered both thresholds of *GDP* per capita and willingness to pay [18,24,29]. Apart from these studies, seven studies undertook an evaluation from a societal perspective [19,22–24,27–29], and four studies utilised the health system perspective [4,18,20,25]. Several studies used the societal perspective and included all vaccination costs, relevant direct medical costs, and gains in quality and length of life without considering who incurred the costs or who received the benefits (Table 2). However, these selected studies reported little about the indirect costs and productivity losses which are significant from the societal perspective.

**Table 2. Characteristics of the selected studies.**

| Author | Study settings | Economic category | Target age cohort | Sex of cohort | Vaccine delivery route | No of doses | Type of model | Threshold | Perspective | Time horizon (year) | Discount rate | Sensitivity analysis | Most sensitive parameter |
|---|---|---|---|---|---|---|---|---|---|---|---|---|---|
| Kiatpongsan et al. [29] | Kenya and Uganda | LMIC | 9 years | Female | NIP | 3 | Static | GDP and WTP | Societal | ns | 3% | One-way | Discount rate |
| Laprise et al. [22] | United States | HI | 9–14 years | Female | NIP | 2 & 3 | Dynamic | WTP | Societal | 100 | 3% | One-way | Vaccine efficacy, screening method, and healthcare costs, vaccine coverage |
| Largeron et al. [23] | Germany | HI | 12–17 years | Female | SHI plans | 2 | Dynamic | WTP | Societal | 100 | 3% | One-way | Discounted rate, vaccine price |
| Mennini et al. [4] | Italy | HI | 12 years | Female | NIP | 2 | Dynamic | WTP | Health system | 100 | 3% | One-way | Vaccine price |
| Mo et al. [18] | China | MI | 12 years | Female | NIP | 3 | Markov | GDP and WTP | Health system | | 3% | One-way | |
| Simms et al. [25] | Australia | HI | 12 years | Female | NIP | 2 | Dynamic | WTP | Health system | 20 | 5% | One-way | Vaccine price and vaccine duration of protection |
| Boiron et al. [17] | Austria | HI | 9 years | Gender-neutral | Universal | 2 | Dynamic | WTP & GDP | Health system | 100 | 3% | One-way | Discount rates and duration of protection |
| Brisson et al. [24] | United States | HI | 9 years | Gender-neutral | Universal | 3 | Dynamic | WTP | Societal | 70 | 3% | One-way | Vaccine price |
| Chesson et al. [27] | United States | HI | Female: 12 to 26 years, and male:12 to 21 years | Gender-neutral | NIP | 3 | Dynamic | WTP | Societal | 100 | 3% | One-way | Vaccine price, Time horizon |
| Chesson et al. [28] | United States | HI | Female:13-18years | female | NIP | 3 | Dynamic | WTP | Societal | 100 | 3% | One-way, Multi-way | Vaccine price |
| Chesson et al. [20] | United States | HI | Female: 12 to 26 years, and male:12 to 21 years | Gender-neutral | NIP | 3 | Dynamic | WTP | Health system | 100 | 3% | One-way, Multi-way | Vaccine price |
| Drolet et al. [19] | Canada | HI | 10 years | Female | NIP | 3 | Static | GDP | Societal | 70 | 3% | One-way, Multi-way | Duration of protection, vaccine efficacy, vaccine price, discount rate |

SHI = Statutory health insurance plans, NIP = National Immunisation Program, WTP = Willingness to pay

## Vaccine coverage

The assumptions on vaccine coverage are significant in influencing the potential impact of *HPV* vaccine on *HPV* related diseases. Four selected studies assumed a vaccination coverage rate of 90% or above [18,22,23,29]. The vaccine coverage might be varied in terms of study settings as well as from a gender point of view. Among the selected studies, three studies considered vaccine coverage rates of 26–60% for females and 25–40% for males [17,27,28], and three studies considered a 46–80% vaccine coverage rate [19,20,25]. Three studies grouped vaccination coverage rate by gender, assumed 25–60% for females and 11–40% for males [24,27,28]. The remaining studies did not specify the vaccination coverage rate [24].

## Vaccine efficacy

Most studies considered a vaccine efficacy rate ranged from 95–100% against *HPV* infections except the study of Simms et al. (2016) [25], which considered a vaccine efficacy rate of only 59%. The study conducted in two East African countries (Kenya and Uganda) used a 100% vaccine efficacy rate in case of *9vHPV* [29]. Most studies (n = 10/12) used a 95% vaccine efficacy rate [4,17,27,28,18–24,26].

## Number of vaccine dose and delivery route

Eight studies used a three-dose schedule of *9vHPV* vaccine. Most studies were conducted in developed countries [18–21,24,27–29] and the other two studies were conducted in low- and middle-income countries (*LMICs*) [18,29]. Further, one study conducted in the United States [21] used both 2- and 3-dose vaccines. Diverse vaccine delivery routes were evidenced across the selected studies. Nine studies used the vaccine delivery route of a national immunisation program for the target population [4,18–21,25,27–29]. Two studies conducted in Austria [17] and United States [24], used a universal immunisation strategy to deliver the vaccine. Only one cost-effectiveness exploration of *9vHPV* vaccine was conducted in Germany [23] and it used a vaccine delivery route through social health insurance.

## Duration of vaccine protection, herd immunity effect, and discounting rate

Most studies (11/12) assumed lifelong vaccine protection while only one study assumed a shorter duration of protection of 20 years [19]. Half of the studies specified herd immunity due to vaccination [17,19,20,24,27,28]. The remaining six studies did not consider the indirect effect of vaccination. Regarding the discount rate, majority of the studies (11/12) used 3% discount rate, while one study considered a 5% discount rate to adjust for future values in terms of economic value and health outcome [25].

## Quality of included studies

The quality scores were assigned using the Consensus Health Economic Criteria (*CHEC*) list, a checklist that can be used to critically evaluate published economic evaluations [30]. Table 3 showed the extent to which the reviewed studies followed the standards of reporting economic evaluations based on the *WHO* guidance [11], Drummond [12] and the Consolidated Health Economic Evaluation Reporting Standards (*CHEERS)* [13]. All studies clearly identified the study question, intervention(s), comparator(s) perspectives, time horizon and discounting rates. Most studies performed sensitivity analyses (11/12; 92%) to assess the robustness of concerned study findings. Most studies clearly described the measurements and the assumptions for measuring the costs (11/12, 92%). The choice of model used was justified in all studies, where dynamic transmission model was adopted to capture herd immunity. The currency and price data were also

**Table 3. Extent to which included studies met standard reporting recommendations.**

| Explained recommendations | Number of studies fulfilling | Percentage (%) |
|---|:---:|:---:|
| Research question or objective clearly stated | 10/12 | 83 |
| Described intervention and comparator | 10/12 | 83 |
| Exploration of effectiveness reported | 11/12 | 92 |
| *Single study-based estimates* | 8/12 | 67 |
| *Synthesis-based estimates* | 10/12 | 83 |
| Assumption of costs and outcomes specified | 11/12 | 92 |
| Currency and price data reported | 12 | 100 |
| Choice of model justified | 12 | 100 |
| Perspective specified | 12 | 100 |
| Time horizon specified | 12 | 100 |
| Discounting rates specified | 12 | 100 |
| Calculated and reported *ICER* or cost-saving | 12 | 100 |
| Sensitivity analysis performed | 11/12 | 92 |
| Conclusions follow from the data reported | 12 | 100 |
| Disclosed funding source(s) | 10/12 | 83 |

reported in all studies. 10 (83%) out 12 studies disclosed the funding sources. However, only 8 studies (67%) reported the measurement of effectiveness from synthesis-based estimates, either through the combination of several randomized trials or the use of systematic reviews.

## Results

Ten studies concluded that their evaluation of *9vHPV* vaccination was found to be cost-effective (Table 4) while the remaining two studies did not find cost-effectiveness [27,28]. Further, five studies exhibited a 'very cost-effective' decision [4,18,19,23,29] and four studies found 'cost-savings' [17,22,24,27]. In the context of high-income countries (e.g., Canada and Austria), introduction of *9vHPV* vaccination was a cost-effective decision to prevent cervical cancer in adolescent girls, as the incremental cost of vaccine was less than US$23-US$47. However, in low and middle-income countries (e.g., Kenya and Uganda), the *ICER* of *9vHPV* vaccine must not be priced over US$8.40-US$9.80 [19,29]. Two *US based* studies concluded that the cost-effectiveness exploration of *9vHPV* vaccine was more likely to be 'cost-saving' regardless of cross-protection assumption [24,27]. Most studies used 'quality-adjusted life year' (*QALYs*) as the unit of measurement. In addition, selected studies explored the cost-effectiveness decision using *WTP* thresholds that depend on country settings. Cost-effectiveness decision differs with country specific vaccine prices. For example, two studies conducted in the *US*, considered two different vaccine prices per dose, US$162.74 and US$174, respectively. However, both studies confirmed that the introduction of *9vHPV* vaccine was not cost-effective. Four studies reported cost-effectiveness of *9vHPV* vaccine for gender-neutral approaches [17,20,24,27] and three studies found it a 'cost-effective' or 'cost-saving' decision [17,24,27]. The remaining eight studies suggested vaccinating girls only. In terms of key drivers of cost-effectiveness, this review identified duration of vaccine protection [17,19,25], time horizon [28], vaccine price [4,19,20,23–25,27,28], healthcare costs [22], vaccine efficacy [19,22], vaccine coverage [19,22] and discounting rates [17,19,23,29] as the most influential parameters.

## Discussion

The *HPV* vaccination is one of the cornerstones of *CC* prevention worldwide. This study explored the cost-effectiveness of *9vHPV* vaccination by reviewing 12 cost-effectiveness

**Table 4. Summary of the results of the selected studies.**

| Author | Vaccine efficacy | Vaccine coverage | Duration of vaccine protection | Herd effect | Vaccine price per dose | Unit of cost-effectiveness | GDP per capita | Incremental cost-effectiveness ration (ICER) | Conclusion or recommendation | Study funder |
|---|---|---|---|---|---|---|---|---|---|---|
| Kiatpongsan et al. [29] | 100% | 100% | Lifetime | No | US$ 90.25 | QALYs | Kenya = $1,349.97, Uganda = $ 674.05 | Very cost-effective if additional cost of 9vHPV vaccine per course ≤ $9.8 in Kenya & ≤ 8.4 in Uganda | Very cost-effective for both countries (Kenya & Uganda) | The Bill and Melinda Gates Foundation |
| Laprise et al. [22] | 95% | 90% | Lifetime | No | US$ 158 | QALYs | | Cost saving to US $ 500 | Cost saving | CDC |
| Largeron et al. [23] | 96% | 90% | Lifetime | No | € 140 | QALYs | £30,000 | € 329 / QALY | Highly cost-effective | Sanofi Pasteur MSD (SPMSD). |
| Mennini et al. [4] | 96% | 90% | lifelong | No | € 80.00 | QALYs | € 40,000 | € 10,463 / QALY | Highly cost-effective | Sanofi Pasteur MSD |
| Mo et al. [18] | 96.7% | 20% | lifetime | No | USD 149.03 | QALYs | USD 23,880 | US$ 5,768 / QALY | Highly cost-effective with screening 1 + 9vHPV,—Cost-effective with screening 2 + 9vHPV | The Japan Society for the Promotion of Sciences, the National Centre for Child Health and Development, and the Chinese Natural Sciences Foundation |
| Simms et al. [25] | 59% | 70% | lifelong | No | ns | QALYs | AUD 30,000 | Cost-effectiveness if the additional cost per dose is US$18–28 | Cost-effective | National Health and Medical Research Council, Australia |
| Boiron et al. [17] | 98% | Female: 60% Male: 40% | Lifelong | Yes | US$ 147.15 | QALYs | US$ 44,767.35 | Cost-saving at vaccine price up to US$ 166.77 | Cost-saving | Sanofi Pasteur MSD |
| Brisson et al. [24] | 95.0% | Not stated | Lifelong | Yes | US$ 158 | QALYs | US$ 48,373.88 | Cost-saving regardless of cross-protection assumptions | Cost-saving if additional cost of vaccine per dose < US$ 13 | CDC, Canadian Research Chair Program |
| Chesson et al. [27] | 95.0% | Female: 25.8% Male: 11.7% | Lifelong | Yes | US$ 162.74 | QALYs | US$ 52,787.03 | Cost-saving regardless of cross-protection assumptions (<$0) | Cost-saving | Not stated |
| Chesson et al. [28] | 95.0% | Female: 46% Male: 25% | Lifelong | Yes | US$ 162.74 | QALYs | US$ 52,787.03 | US$ 111,446 / QALY | Not cost-effective | CDC, Canada Research Chair Program, Canadian Institute for Health Research |
| Chesson et al. [20] | 95.0% | 46% | Lifelong | Yes | US$ 174 | QALYs | US$ 52,787.03 | US$ 228,800 / QALY | Not cost-effective | Not stated |
| Drolet et al. [19] | 95.0% | 80% | 20years | Yes | US$ 90.25 | QALYs | US$ 50,440.44 | US$ 11,593 /QALY | Very cost-effective if additional cost of vaccine per dose ≤ US$ 22.80 | Canadian Research Chair Program |

evaluations in order to inform and expand knowledge on the cost-effectiveness of *9vHPV* vaccines. Most studies were conducted from a developed country perspective and two studies were performed from a *LMIC* perspective. However, a higher incidence of cervical cancer in *LMICs* is a serious public health concern, which warrants more evidence for effective decision

making [31]. The economic viability of gender-neutral *9vHPV* vaccination was confirmed by three studies [17,24,27]. Cost-effectiveness exploration depends on the coverage of vaccination from the perspective of gender. For example, if the vaccine coverage for female recipients is 80% or above, the majority of the anogenital *CC* including vulvar cancers, invasive vaginal carcinomas cancers in females could be prevented. As a result, introduction of *9vHPV* vaccination for boys is relatively less important compared with girls as high economic costs are involved without additional benefits gained, both from the societal and health system perspectives. Therefore, achieving optimal coverage of vaccination in females should remain a priority. This is of primary significance for *LMICs* settings since it is more effective and economically viable to prevent *CC* in females. However, it is also important to note that past studies paid little attention to the broader benefits of vaccination among male cohorts to prevent penile, anal, and oropharyngeal cancers. Exclusion of these diseases related to males may undermine the effectiveness of reducing *CC*. Gender-neutral vaccination might have several benefits including herd protection for boys. Moreover, it may provide indirect protection to unvaccinated women and direct protection to homosexual men. Therefore, this vaccination strategy should be further considered in country-level immunization programs by underlining other parameters including disease burden, sexual behaviour in a country (e.g., homosexual intercourse), equity, budget impact, and affordability.

Despite different methodologies and various assumptions, most studies were consistent in their conclusion that multiple age cohort vaccination was economically viable. Nevertheless, there was an upper age limit at which *HPV* vaccination was no longer cost-effective, and should be interpreted cautiously as several studies evaluated the cost-effectiveness in a single age range only and did not compare to the next age range in a progressive manner. Subsequently, this could result in an overestimation of the cut-off age range for vaccination. The protection duration from vaccination has a large impact on the cost-effectiveness of multi-cohort vaccination, with most studies assuming life-long protection. Therefore, the use of *ICERs* based on the conventional evaluation of 10-year protection may be more representative of real-life effectiveness rather than the use of *ICER* based on lifetime protection. The cost-effectiveness of *HPV* vaccination is also dependent upon the levels of vaccine coverage, compliance, and vaccine price.

Most studies presumed a high rate of vaccination coverage, e.g., assumed that 70% of the target population will receive full doses of vaccination. However, not everyone completed full doses (i.e., two or three doses) within the recommended time frame. Therefore, cost-effectiveness evaluation may underestimate or overestimate the actual costs and benefits. The analytical model outcomes in terms of herd immunity is only hypothetical unless the coverage level increases among the study cohort. Further, it is also indeterminate how non-compliance may consequently influence vaccine efficacy, effectiveness and duration of protection. Model input assumptions regarding the *9vHPV* vaccine price also influence the observed cost-effectiveness outcomes. Prices for *9vHPV* vaccine are currently not specified, particularly, in lower-income countries. Hence, the cost-effectiveness of *9vHPV* vaccine is still indeterminate and there is no exclusive evidence of greater cost-effectiveness than the older licensed *HPV* vaccines.

Therefore, once the 9-valent vaccine price is fixed, including support by the *GAVI* vaccine-alliance, reassessment of cost-effectiveness of *9vHPV* vaccine is necessary. Another model input assumption that may influence the cost-effectiveness is the inclusion or exclusion of herd immunity effects based on the type of model acceptance. Two studies [19,29] constituted the static model as an analytical exploration which did not confirm herd immunity effects. Generally, the cost-effectiveness evaluations of *HPV* vaccine should use a dynamic model for exploration because economic evaluations for primary prevention strategy should be determined by societal benefits (e.g., indirect impacts on population not immunised) rather than

individual demands [32]. However, the application of a static model in these two studies may underestimate or overestimate the benefits of vaccination. If an *HPV* vaccination program is exhibited to be cost-effective considering a static model for analytical exploration, it is anticipated to be even very cost-effective when a dynamic model is considered [32].

There are several types of cost-effectiveness threshold. The majority of the studies used the cost-effectiveness demand side-threshold (e.g. willingness-to-pay). In health-related explorations, a willingness-to-pay threshold signifies an evaluation of what a consumer of health care might be prepared to pay for the health benefit–given other competing demands on that consumer's resources. There are also supply-side thresholds that resource allocation mechanism takes into account. For example, estimates of health status are predetermined since when an insurance company or other provider spends some of its available budget on a new intervention it is therefore required to decrease its funding of previous interventions. In considering the choice of the type of cost-effectiveness threshold to use, the concept of opportunity cost may be the one most relevant to providers who are primarily concerned with using available resources to maximise improvements in health status. In response to the implementation of a new intervention, decision-makers need estimates of both the health that might be gained elsewhere through the alternative use of the resources needed for the new intervention and the health that is likely to be lost if the new intervention is not used.

This review has some limitations. The cost-effectiveness evaluation based on *GDP* based thresholds of 1–3 times of *GDP* per capita might be misleading for country-level decision making due to a lack of country specific thresholds [33]. It is uncertain whether this threshold truly reflects the country's affordability or societal willingness to pay for additional health gains. Additionally, *GDP* is originally intended to measure the experience of people residing in urban areas and thus it may not actually reflect the experience of the entire population in a country, especially those living in rural areas. Apart from an economic standpoint, other factors should be considered for the national immunization program, such as budget availability, political issues, cultural influences and availability of healthcare workforce.

## Conclusions

There are a limited number of studies that showed conclusive evidence of cost-effectiveness of the *9vHPV* vaccine. The inclusion of adolescent males in *HPV* vaccination programs is cost-effective subject to vaccine price or coverage of females being low and *HPV*-associated male diseases are taken into account. Multiple age cohort vaccination strategy is likely to be cost-effective in the age range of 9–14 years, but the upper age limit at which *HPV* vaccination is no longer cost-effective requires further investigation. Vaccine coverage, price, duration of protection and discount rates are important parameters for considering the uptake of *HPV* vaccination. Nonetheless, the present study findings may serve as useful evidence for health policy-makers and healthcare providers in taking decision about *HPV* national immunization programs using the new *9vHPV* vaccine or inclusion of adolescent boys' for vaccination or extending the age of immunization.

## Supporting information

**S1 Checklist.**
(DOC)

**S1 Appendix.**
(DOCX)

**S1 Data.**
(DTA)

## Acknowledgments

We would like to gratefully acknowledge the study participants and reviewers and editors of our manuscript.

## Author Contributions

**Conceptualization:** Rashidul Alam Mahumud.

**Data curation:** Rashidul Alam Mahumud, Syed Afroz Keramat.

**Formal analysis:** Rashidul Alam Mahumud.

**Funding acquisition:** Rashidul Alam Mahumud.

**Investigation:** Rashidul Alam Mahumud, Syed Afroz Keramat.

**Methodology:** Rashidul Alam Mahumud.

**Project administration:** Rashidul Alam Mahumud, Jeff Dunn, Jeff Gow.

**Resources:** Rashidul Alam Mahumud, Syed Afroz Keramat, Gail M. Ormsby.

**Software:** Rashidul Alam Mahumud.

**Supervision:** Khorshed Alam, Jeff Dunn, Jeff Gow.

**Validation:** Rashidul Alam Mahumud, Khorshed Alam, Syed Afroz Keramat, Gail M. Ormsby, Jeff Dunn, Jeff Gow.

**Visualization:** Rashidul Alam Mahumud, Khorshed Alam, Gail M. Ormsby, Jeff Dunn, Jeff Gow.

**Writing – original draft:** Rashidul Alam Mahumud, Khorshed Alam, Syed Afroz Keramat, Gail M. Ormsby, Jeff Dunn, Jeff Gow.

**Writing – review & editing:** Rashidul Alam Mahumud, Khorshed Alam, Syed Afroz Keramat, Gail M. Ormsby, Jeff Dunn, Jeff Gow.

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
