## [Decision Letter · Decision Letter 0]

8 Apr 2020

PONE-D-19-28575

Mapping of cost-effectiveness evaluations of the 9-Valent human papillomavirus (HPV) vaccine: Evidence from a systematic review

PLOS ONE

Dear Dr Alam Mahumud,

Thank you for submitting your manuscript to PLOS ONE. After careful consideration, we feel that it has merit but does not fully meet PLOS ONE’s publication criteria as it currently stands. Therefore, we invite you to submit a revised version of the manuscript that addresses the points raised during the review process.

We would appreciate receiving your revised manuscript by June 8th, 2020. To enhance the reproducibility of your results, we recommend that if applicable you deposit your laboratory protocols in protocols.io, where a protocol can be assigned its own identifier (DOI) such that it can be cited independently in the future. For instructions see: http://journals.plos.org/plosone/s/submission-guidelines#loc-laboratory-protocols

We look forward to receiving your revised manuscript.

Kind regards,

Magdalena Grce, PhD

Academic Editor

PLOS ONE

2. At this time, we ask that you please provide the full search strategy and full search terms for at least one database used as Supplementary information. In addition, please state whether a third database was included in your search strategy.

Thank you for including the statement that "Searches were performed until 31 July 2019". Please revise this statement to clarify whether all databases were searched from inception, or if there were any limits placed on the publication dates in your search. In addition, please include this statement in the Methods (rather solely in the abstract).

"The study is part of the first author’s PhD research works. The PhD program was funded by the University of Southern Queensland, Australia."

5. Please amend your list of authors on the manuscript to ensure that each author is linked to an affiliation. Authors’ affiliations should reflect the institution where the work was done (if authors moved subsequently, you can also list the new affiliation stating “current affiliation:….” as necessary).

6. Please include your tables as part of your main manuscript and remove the individual files. Please note that supplementary tables (should remain/ be uploaded) as separate "supporting information" files. 

Reviewers' comments:

Reviewer's Responses to Questions

**Comments to the Author**

1. Is the manuscript technically sound, and do the data support the conclusions?

Reviewer #1: Yes

Reviewer #2: Partly

2. Has the statistical analysis been performed appropriately and rigorously? 

Reviewer #1: Yes

Reviewer #2: N/A

3. Have the authors made all data underlying the findings in their manuscript fully available?

Reviewer #1: Yes

Reviewer #2: No

4. Is the manuscript presented in an intelligible fashion and written in standard English?

Reviewer #1: Yes

Reviewer #2: Yes

5. Review Comments to the Author

Reviewer #1: The manuscript is scientifically sound and justice has been made into the writing only minor changes have to be made.

Reviewer #2: This is a useful and interesting paper that is generally well presented.

There are a number of areas in which it could however be improved.

Major issue:

Discuss the role choice of comparator has in the reported ICER and report same in the Tables.

Minor issues

First, clarify why only Scopus and Pubmed were searched.

Second, ensure the exclusion criteria detailed in study selection align with those detailed in the study characteristics subsection (or back reference the earlier mentioned criteria to avoid confusion as an alternative configuration).

Third, page 5 line 17 - what constitutes reasonable and how was this determined?

Fourth, page 5, line 16 ff. identify by initial which authors did what.

Page 5, line 20 reference CHEC as the risk of bias tool

Fifth, page 6 line 19-22. I'm unclear what the text related to CET adds and indeed may serve to confuse matters.

Sixth, page 7 line 6-10, clarify the distinction between national immunization program and universal immunization strategy and in what sense is social health insurance a delivery as opposed to a funding mechanism?

Page 9, line 24 ff. majority suggests some did not report perspective etc. whereas 100% means "all".

Page 10, line 4, find an alternative sub-heading title as the preceding material also cover results.

Page 10, line 15 - QALY is an outcome measure not cost-effectiveness

Page 11, line 3, the study examined 2 LMICs not "an LMIC".

Page 12 line 1-2, ICER is also heavily dependent on the choice of comparator - see above.

Page 12, line 6, under/over-estimate costs and benefits

Page 12, line 14/15 - the assumption of lowest price of 9 valent vaccine. I was unclear as to what exactly the authors meant here. Is this context specific?

Page 12/13 Paragraph beginning "A cost-effectiveness threshold....if the new intervention is not used." Suggest you remove this as it adds little and may serve to confuse.

Page 13, line 13 - the suggestion that country specific GDPs lack country specificity seems odd.

Page 13, line 22/23 - the opening line of the conclusion seems overly pessimistic regarding the potential cost-effectiveness of the vaccine.

Tables - carefully proof - e.g. Table 4 "vaccine prince"

6. PLOS authors have the option to publish the peer review history of their article (what does this mean?). If published, this will include your full peer review and any attached files.

Reviewer #1: Yes: Akim Tafadzwa Lukwa

Reviewer #2: No

---

## [Author Response · Author response to Decision Letter 0]

4 May 2020

Cover Letter

Date: April 28, 2020 

Dear Editor and Reviewer’s, 

Thank you for giving us an opportunity to revise our manuscript entitled “Cost-effectiveness evaluations of the 9-Valent human papillomavirus (HPV) vaccine: Evidence from a systematic review”. We found the reviewers’ comments/feedback very helpful in improving the manuscript and we have revised the manuscript accordingly. Please find attached the revised manuscript. We declare that all authors have no conflicts of interest. The manuscript has not been published in any other journal. Our point-by-point comments on the suggested revisions are below. 

Best regards,

Rashidul Alam Mahumud (corresponding author)

PhD Candidate, MPH, MSc

On behalf of all of the co-authors

Health Economics Research, 

Health Systems and Population Studies Division, 

International Centre for Diarrhoeal Disease Research, Bangladesh (icddr,b), 

Dhaka-1212, Bangladesh.

Requirement 1. Please ensure that your manuscript meets PLOS ONE's style requirements, including those for file naming. The PLOS ONE style templates can be found at http://www.plosone.org/attachments/PLOSOne_formatting_sample_main_body.pdf and http://www.plosone.org/attachments/PLOSOne_formatting_sample_title_authors_affiliations.pdf

 Author’s response: The manuscript has been revised according to the PLoS ONE’s requirements. We hope that the revisions made have met PLoS ONE’s style requirements. 

Requirement 2. At this time, we ask that you please provide the full search strategy and full search terms for at least one database used as Supplementary information. In addition, please state whether a third database was included in your search strategy. Thank you for including the statement that "Searches were performed until 31 July 2019". Please revise this statement to clarify whether all databases were searched from inception, or if there were any limits placed on the publication dates in your search. In addition, please include this statement in the Methods (rather solely in the abstract).

Author’s response: We have already uploaded the full search terms for PubMed and Scopus databases. The text has been updated now. A literature search covering a period of January 2000 to 31 July 2019 was conducted in PubMed and Scopus bibliographic databases. We searched for articles with no language restrictions.

 Requirement 3. In your Data Availability statement, you have not specified where the minimal data set underlying the results described in your manuscript can be found. PLOS defines a study's minimal data set as the underlying data used to reach the conclusions drawn in the manuscript and any additional data required to replicate the reported study findings in their entirety. All PLOS journals require that the minimal data set be made fully available. For more information about our data policy, please see http://journals.plos.org/plosone/s/data-availability.

Author’s response: We have uploaded underlying data set as a supporting information files.

 Requirement 4. Important: If there are ethical or legal restrictions to sharing your data publicly, please explain these restrictions in detail. Please see our guidelines for more information on what we consider unacceptable restrictions to publicly sharing data: http://journals.plos.org/plosone/s/data-availability#loc-unacceptable-data-access-restrictions. Note that it is not acceptable for the authors to be the sole named individuals responsible for ensuring data access. We will update your Data Availability statement to reflect the information you provide in your cover letter.

Author’s response: We have updated the data availability statement in the submission system. 

Requirement 5. Thank you for stating the following in the Acknowledgments Section of your manuscript: "The study is part of the first author’s PhD research works. The PhD program was funded by the University of Southern Queensland, Australia." We note that you have provided funding information that is not currently declared in your Funding Statement. However, funding information should not appear in the Acknowledgments section or other areas of your manuscript. We will only publish funding information present in the Funding Statement section of the online submission form. Please remove any funding-related text from the manuscript and let us know how you would like to update your Funding Statement. Currently, your Funding Statement reads as follows: "The author(s) received no specific funding for this work."

Author’s response: We have revised the acknowledgement section to remove the sentence “The PhD program was funded by the University of Southern Queensland, Australia”. Please see the revised manuscript. 

Requirement 6. Please amend your list of authors on the manuscript to ensure that each author is linked to an affiliation. Authors’ affiliations should reflect the institution where the work was done (if authors moved subsequently, you can also list the new affiliation stating “current affiliation:.” as necessary).

Author’s response: Authors’ affiliations have been corrected now. 

Requirement 7. Please include your tables as part of your main manuscript and remove the individual files. Please note that supplementary tables (should remain/ be uploaded) as separate "supporting information" files.

Author’s response: We have included all tables in the revised main manuscript. The individual Tables have been removed from the submission system. 

Response to the reviewer’s comments

Response to Reviewer #1 comments:

Comment-1: Reviewer #1: The manuscript is scientifically sound and justice has been made into the writing only minor changes have to be made.

Response: Authors express gratitude to the reviewer for positive comments regarding the manuscript. The manuscript has been revised now. We hope that our revisions meet this journal’s standards.

Response to Reviewer #2 comments:

Comment 1. This is a useful and interesting paper that is generally well presented.

Author’s Response: Authors express gratitude to the reviewer for their appreciation.

Comment 2. There are a number of areas in which it could however be improved.

Author’s Response: The manuscript has been revised extensively now. We hope that our revisions meet this journal’s standards. Please see the revised manuscript.

Comment 3. Discuss the role choice of comparator has in the reported ICER and report same in the Tables.

Author’s Response: The text has been revised now to remove the term ‘comparator’ from the revised version of the manuscript. Please see page 10 (line 1).

Comment 4. First, clarify why only Scopus and Pubmed were searched.

Author’s Response: We would like to thank the reviewer for asking this clarification. The present study was conducted without any financial supports. This study was a part of the first author’s PhD research works, which was explained in the acknowledgement section. We have considered (e.g., Scopus and PubMed) search engines due to the limited resources and time constraints. It would have been scientifically more robust if the authors were able to consider other search engines. However, most of the publishers in medical and public health journals are indexed in Scopus (14,000 of Journals) and PubMed (30,000 of Journals). Databases, such as Scopus and PubMed, utilise search interfaces that offer a greater variety of advanced features. Furthermore, PubMed delivers a publicly available search interface for MEDLINE as well as other NLM resources, making it the premier source for biomedical literature and one of the most widely accessible resources in the world. Health sciences practitioners, researchers, faculty, and students have been repeatedly reported PubMed and MEDLINE. 

Comment 5. Second, ensure the exclusion criteria detailed in study selection align with those detailed in the study characteristics subsection (or back reference the earlier mentioned criteria to avoid confusion as an alternative configuration).

Author’s Response: We have revised the study selection section to remove the exclusion criteria. We hope that the revisions made addressed the concerns raised by the reviewer. Please see page 5 (lines 7-9). 

Comment 6. Third, page 5 line 17 - what constitutes reasonable and how was this determined?

Author’s Response: The text has been removed from the revised version of the manuscript. Please see page 5 (lines 21-23).

Comment 7. Fourth, page 5, line 16 ff. identify by initial which authors did what.

Author’s Response: The sentence has been revised now. Please see page 5 (lines 21-23).

Comment 8. Page 5, line 20 reference CHEC as the risk of bias tool

Author’s Response: The text has been revised now. Please page 12 (lines 1-2).

Comment 9. Fifth, page 6 line 19-22. I'm unclear what the text related to CET adds and indeed may serve to confuse matters.

Author’s Response: Thank you for the reviewer’s valuable concern. The sentence has been removed now. Please see on page 6 (lines 24-27).

Comment 10. Sixth, page 7 line 6-10, clarify the distinction between national immunization program and universal immunization strategy and in what sense is social health insurance a delivery as opposed to a funding mechanism?

Author’s Response: Thank you. We have reported the vaccine delivery strategies based on the original studies. The distinctive nature of the national immunization program and universal immunization strategy is out of scope of this study. 

Comment 11. Page 9, line 24 ff. majority suggests some did not report perspective etc. whereas 100% means "all".

Author’s Response: The text has been revised now accordingly. Please see page 12 (lines 8-15).

Comment 12. Page 10, line 4, find an alternative sub-heading title as the preceding material also cover results.

Author’s Response: Thank for the reviewer’s advice. We explained the study characteristics which was captured in the methods and materials sections. In the results section, we have considered the main findings in terms of economic viability of 9-valent HPV vaccination and associated dominating model parameters. 

Comment 13. Page 10, line 15 - QALY is an outcome measure not cost-effectiveness

Author’s Response: We agree with the observation. The text has been revised now (please see page 13 (lines 13-14).

Comment 14. Page 11, line 3, the study examined 2 LMICs not "an LMIC".

Author’s Response: Corrected. Please see page 16 (lines 6-7). 

Comment 15. Page 12 line 1-2, ICER is also heavily dependent on the choice of comparator - see above.

Author’s Response: Thank you. The text has been removed now. Please see page 18 (line 3-4).

Comment 16. Page 12, line 6, under/over-estimate costs and benefits

Author’s Response: Corrected. Please see page 17 (line 14). 

Comment 17. Page 12, line 14/15 - the assumption of lowest price of 9 valent vaccine. I was unclear as to what exactly the authors meant here. Is this context specific?

Author’s Response: Thank you for the reviewer’s concern. The sentence has been removed from the revised version of the manuscript. Please see page 17 (lines 22-23).

Comment 18. Page 12/13 Paragraph beginning "A cost-effectiveness threshold....if the new intervention is not used." Suggest you remove this as it adds little and may serve to confuse.

Author’s Response: The text has been removed now. Please see page 18 (lines 3-4).

Comment 19. Page 13, line 13 - the suggestion that country specific GDPs lack country specificity seems odd.

Author’s Response: Corrected and Revised. Please see page 18 (lines 19-22). 

Comment 20. Page 13, line 22/23 - the opening line of the conclusion seems overly pessimistic regarding the potential cost-effectiveness of the vaccine.

Author’s Response: Corrected and Revised. Please see page 18 (line 30-32). 

Comment 21. Tables - carefully proof - e.g. Table 4 "vaccine prince"

Author’s Response: Thank you for the reviewer’s valuable concern. The text has been corrected now. Please see Table 4.

---

## [Editor Report · Decision Letter 1]

7 May 2020

Cost-effectiveness evaluations of the 9-Valent human papillomavirus (HPV) vaccine: Evidence from a systematic review

PONE-D-19-28575R1

Dear Dr. Mahumud,

We are pleased to inform you that your manuscript has been judged scientifically suitable for publication and will be formally accepted for publication once it complies with all outstanding technical requirements.

With kind regards,

Magdalena Grce, PhD

Academic Editor

PLOS ONE
---

## [Editor Report · Acceptance letter]

11 May 2020

PONE-D-19-28575R1 

Cost-effectiveness evaluations of the 9-Valent human papillomavirus (HPV) vaccine: Evidence from a systematic review 

Dear Dr. Mahumud:

I am pleased to inform you that your manuscript has been deemed suitable for publication in PLOS ONE. Congratulations! Your manuscript is now with our production department. 

With kind regards,

on behalf of

Dr. Magdalena Grce 

Academic Editor

PLOS ONE